# Mineral-Solubilizing Soil Bacteria Permanently Green Rocky Slopes by Enhancing Soil Adhesion to the Surface of Rocky Slopes

**Lingjian Wang** [1][ID]**, Xinggang Tang** [2]**, Xin Liu** [1] **and Jinchi Zhang** [1,*]

1. Co-Innovation Center for Sustainable Forestry in Southern China,
   Jiangsu Province Key Laboratory of Soil and Water Conservation and Ecological Restoration,
   Nanjing Forestry University, 159 Longpan Road, Nanjing 210037, China
2. Jiangxi Institute of Land Space Survey and Planning, 66 Unity Road, Xihu District, Nanchang 330025, China
* Correspondence: zhang8811@njfu.edu.cn

**Abstract:** Rocky slopes are vulnerable to landslides and mudslides, which pose a major threat to human life and property. Research is being conducted to improve the adhesion between soil and minerals by mineral-solubilizing bacteria to manage slopes scientifically and develop novel methods for slope greening. From the soil of Nanjing Mufu Mountain's weathered rock walls, we isolated various soil mineral-solubilizing soil bacteria. During the soil bacterial solubilization test, we discovered that some soil bacteria could enhance the adherence of soil to minerals; therefore, we selected three soil bacteria (NL-7, NL-8, and NL-11) with higher performance for further investigation. Controlled experiments were used to investigate the effects of soil bacteria on soil characteristics (soil moisture content, soil pH, and soil exchangeable metal content) and soil adhesion to minerals. According to the findings, soil bacteria can improve the soil's adhesion to minerals, improve the soil's capacity to hold water, regulate soil pH, and solubilize and release exchangeable calcium, magnesium, sodium, and potassium ions. A structural equation modeling analysis was performed to thoroughly examine the relationship between soil characteristics and soil adherence to minerals. The analysis findings showed that soil moisture had the greatest total and direct positive impact on soil adherence to minerals. The most significant indirect impact of soil pH on soil adhesion to minerals is mainly caused by the exchangeable sodium and magnesium ions. Additionally, soil exchangeable sodium ions can only indirectly affect the adhesion of soil to minerals, which is accomplished by controlling soil exchangeable magnesium ions. Therefore, mineral-solubilizing soil bacteria primarily work by enhancing the soil's water retention capacity to improve the soil's adherence to minerals. Our study on the effect of mineral-solubilizing bacteria on the adhesion of soil and minerals demonstrates the significant potential of mineral-solubilizing bacteria in spray seeding greening, which will provide data and theoretical support for the formation, application, and promotion of mineral-solubilizing bacteria greening methods and gradually form a new set of scientific and efficient greening methods with Chinese characteristics.

**Keywords:** scientific slope management; mineral-solubilizing bacteria greening method; adhesion of soil to minerals; structural equation modeling

## 1. Introduction

Mining contributes significantly to global, social, and economic development. However, it also causes many negative ecological and environmental effects [1,2], including eradicating natural flora, irreversible deterioration of soil quality, and the subsequent appearance of many slopes of bare rock [3]. There is a risk to human life and property when bare rock slopes deteriorate in the microclimate, resulting in landslides and mudslides [4]. Therefore, experts and scholars are increasingly focusing on the ecological restoration of bare rock slopes. Currently, the technologies for ecological restoration of bare rock slopes

domestically and internationally are generally composed of spray greening, vegetation blanket, green bag, crop trough greening, planting hole greening, and thick substrate technologies [5–7]. The most widely used technology is spray greening. Still, it is easy to peel off because it is difficult to maintain the greening effect for a long time and the substrate is poorly adapted to the building surface of rocky slopes. It may be necessary to strengthen the adherence of the sprayed substrate to the surface of the rocky slope construction in order to maintain the stability of the slope ecology and achieve permanent greening of the rocky slopes.

Currently, most soil adhesion research is focused on agriculture [8] to enhance the efficiency of agricultural machinery operations by lowering the adhesion and resistance of soil to tools through morphological modifications, surface engineering, and bionics [9–11]. The characteristics of the solid surface, soil characteristics, and environmental factors all play a major role in the adherence of soil to solids [12]. Soil parameters are typically correlated with the bulk density, moisture content, soil solution quality, and mineral composition of the soil. Numerous studies have demonstrated that soil moisture content plays a significant role in determining the adherence of soil to solids [12–14]. Soil adhesion is greatest when the soil water content is between the liquid limit and the soil plastic limit. Additionally, the frictional, geometrical, and other characteristics of the solid surface all affect the adherence of soil to solids. Several studies have shown that solid materials with high surface free energy and good hydrophilicity strongly adhere [15–17]. The adherence of soil to solids can also be influenced by environmental factors such as process pressure, temperature, etc. There is little literature on the role of soil adhesion in ecological restoration, and its significance in this process has yet to be fully understood [18,19].

It is well known that using bacteria to restore the environment is both economical and eco-friendly. Many bacteria have been shown to promote plant growth [3,5,12,20–22] by enhancing soil structure and nutrient cycling [23–25]. According to Ortiz et al., bacteria also help to mitigate the impacts of salinity and drought stress [26,27]. Additionally, by utilizing the metabolic and antagonistic characteristics of bacteria, efforts have been made to reduce the prevalence of plant pests and diseases in slope management [17,28–32]. In previous studies [33–38], we isolated many bacteria from the soil along weathered dolomite rock walls in the Nanjing Mufu Mountains. These bacteria can alter soil nutrient levels, plant growth characteristics, and the solubilization and release of metal ions from minerals. There are no studies on the impact of soil characteristics on soil adhesion to minerals under the action of bacteria, and it has been reported that bacteria affect soil adhesion to minerals.

Therefore, based on previous studies' findings, we conducted a comprehensive analysis of a series of soil bacteria's solubilization, plant growth promotion, and soil nutrient regulation abilities. Based on this, three indigenous dominant soil bacteria with better-combined effects and tolerance were selected for this study's controlled indoor and outdoor experiments. The objectives were to (1) determine how mineral-solubilizing soil bacteria affected soil adhesion to minerals; (2) look into how they affected soil characteristics; and (3) determine how soil characteristics affected soil adhesion to minerals when mineral-solubilizing soil bacteria were present.

The findings of this study will investigate new areas of soil adhesion research and add to the existing information on the effects of soil bacteria on soil adhesion to minerals. Findings that suggest that soil bacteria can enhance the adhesion of soil to minerals may offer a new direction for the advancement of slope greening methods. Furthermore, such a finding could potentially resolve the issue of traditional greening effectiveness being difficult to maintain over the long term and develop a new spray greening method adapted to the unique climate and environment of China.

## 2. Materials and Methods

### 2.1. Sample Preparation

The strains (NL-7, NL-8, and NL-11) used in this experiment were selected through comprehensive analysis from numerous soil strains isolated from the weathered rock wall

soil of Nanjing Mufu Mountain. The three bacterial strains were identified by 16S rRNA as *Bacillus megaterium* [39], *Bacillus cereus*, and *Bacillus thuringiensis* [40] (preserved in the China Center for Type Culture Collection (CCTCC): M2012452 and CCTCC: M2012453). The well-preserved strains (NL-7, NL-8, and NL-11) were inoculated on Nutrient Agar medium (peptone 10.0 g/L, beef extract 3.0 g/L, sodium chloride 5.0 g/L, and agar 18.0 g/L) and cultured at 30 °C for 24–48 h. The more developed colonies of each strain were selected, transferred to Nutrient Broth medium, shaken at 30 °C for 24 h, filtered, and then placed in sterile water to make a bacterial solution (adjusted to a concentration of $5 \times 10^8$ cfu/mL).

The soil and rock samples used were: (1) Soil sample #1: Mufu Mountain soil samples were collected, air dried, filtered through a 100 mesh (150 μm) screen, and autoclaved before use; (2) Soil sample #2: Soil sample #1 sieved through a 1340 mesh (10 μm) screen, and the selected soil particles were autoclaved before use; (3) Mineral sample #1: Mineral samples from carbonate mines were extracted and progressively sanded with 16, 40, 100, 150, 220, 400, 600, 800, 1200, and 1500 grain sandpaper before being cut into 5 cm × 5 cm × 1 cm (L × W × H) and autoclaved before use; (4) Mineral samples #2: Mineral samples from carbonate mines were extracted and progressively sanded with 16, 40, 100, 150, 220, 400, 600, 800, 1200, and 1500 grain sandpaper before being cut into 20 cm × 20 cm × 1 cm (L × W × H) and autoclaved before use.

### 2.2. Experimental Design

### 2.2.1. Indoor Experiment

Equal volumes of the bacterial solutions NL-7, NL-8, and NL-11 were sprayed onto the surface of minerals (mineral sample #1), and equal amounts of soil particles (soil sample #2), treated with sterile water as a control, were sprayed over them. The adhesion capabilities between soil particles and minerals were assessed after seven days of aseptic management.

### 2.2.2. Pot Experiment

Equal amounts of each bacterial solution (NL-7, NL-8, and NL-11) were thoroughly mixed with equal amounts of soil sample #1, while sterile water treatment served as the control group. The pull ring was attached to the outside of the mineral sample above, and they were filled in between two horizontally positioned mineral samples #2 (as shown in Figure 1). Soil moisture sensors (EC-5) were connected to the METER Em50/G 5-channel data collector where they were placed at the interface of soil and minerals. Blank controls and treatments using additional strains NL-7, NL-8, and NL-11 were labeled as P1, P2, P3, and CK, respectively. Each treatment was given an initial moisture content of 0.3 $m^3/m^3$ (*v/v*) and was incubated aseptically for 28 days. Moisture content was measured every hour, and adhesion was measured every four days. Every four days, soil samples from the mineral surface were collected to determine the soil's pH and exchangeable calcium, magnesium, sodium, and potassium.

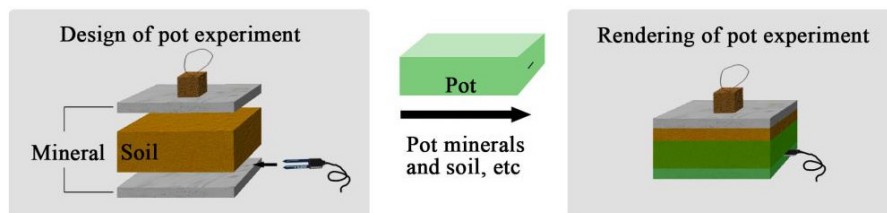

**Figure 1.** The design of the pot experiment. #2 is filled with a soil sample between two mineral samples. The upper mineral sample has a pull ring on the outer side. The soil moisture sensor (EC-5) is inserted at the interface between the lower mineral sample and the soil sample. The samples are placed in a pot and aseptically incubated.

### 2.3. Evaluating the Adhesion of Soil Particles to Minerals

In the indoor experiments, Atomic Force Microscopy (AFM) was used to assess the soil particles' adherence to minerals. AFM, scanning force microscopy, was used to see how the probe and the sample interacted. It was composed of five components: a microcantilever with a tip, a piezoelectric scanner, a laser, a photoelectric detector, and a feedback control system. AFM scans the samples measured by a fixed tip on the micro-cantilever, using the optical or tunnel current detection method to retrieve the micro-cantilever position changes and gather information such as the sample surface topography, mechanical characteristics, and other details. The near sample stage and the withdrawal process stage are included in each tip scan.

The experiment used the Dimension Icon/Multimode 8 AFM developed by Bruker Company. The resolution was 0.15 nm in the lateral direction and 0.04 nm in the vertical direction, with the maximum scanning range being 90 μm × 90 μm × 10 μm. The tests were conducted at 25 ± 0.5 °C and relative humidity of 45 ± 2%. The AFM was set to contact mode in this experiment. Probes were used with voltages of 0 V, 0.1 V, 0.3 V, 0.5 V, 0.7 V, and 0.9 V. Fifty soil particle samples were randomly selected from the mineral sample's surface to observe the state of change for a specific voltage and count the number of displacements that occurred. Each treatment was performed in triplicate (Figure 2) [41,42].

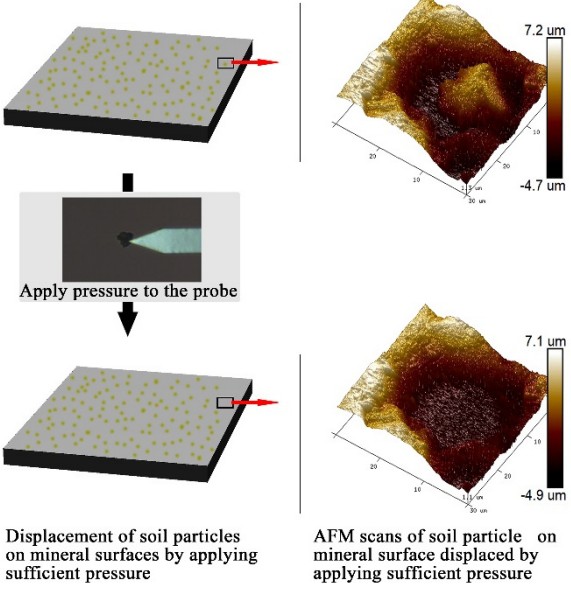

**Figure 2.** Schematic diagram of a probe approaching uniformly close soil particles on a mineral surface. Changes in the morphology of the mineral surface was observed and recorded by tapping the soil particles with an atomic force microscope probe.

### 2.4. Measurements of Soil Characteristics and Adhesion of Soil to Minerals

Plotting experiments measure the soil's adherence to minerals and other soil characteristics. The adhesion of soil to the mineral was measured as the difference between the maximum value exhibited by the tensiometer when the mineral sample is moved, and the value after the mineral is detached from the soil surface (Figure 1) using an ELECALL tensiometer. Soil moisture content was determined using an Em50/G 5−channel data collector. Atomic absorption spectroscopy was used to calculate the exchangeable calcium and magnesium in the soil, while ammonium acetate flame spectrophotometry was used to calculate the exchangeable sodium and potassium.

### 2.5. Statistical Evaluation

Nanoscope analysis software, version 1.8, was used to analyze scanned images. The IBM statistical package for social sciences software, version 20.0, was used for statistical

analysis of the data. Significant differences between different treatments were identified using one-way analysis of variance and least significant difference tests (LSD, $p < 0.05$). Data were plotted using Origin 2021. Images were created with Photoshop CS6.

## 3. Results

### 3.1. The Adhesion of Soil to Minerals

Figure 3 displays the results of the indoor experiment on the adherence of soil to minerals. The number of soil particles that were gradually displaced increased with the applied voltage, with the voltage at which the peak occurred being significantly larger in the added bacteria group than in the control group (CK) (Figure 3). The average voltages of soil particles displaced on the mineral surface in the P1, P2, and P3 treatments were 0.44 V, 0.38 V, and 0.53 V, respectively, which increased by 51.72%, 31.03%, and 60.49% compared to the CK.

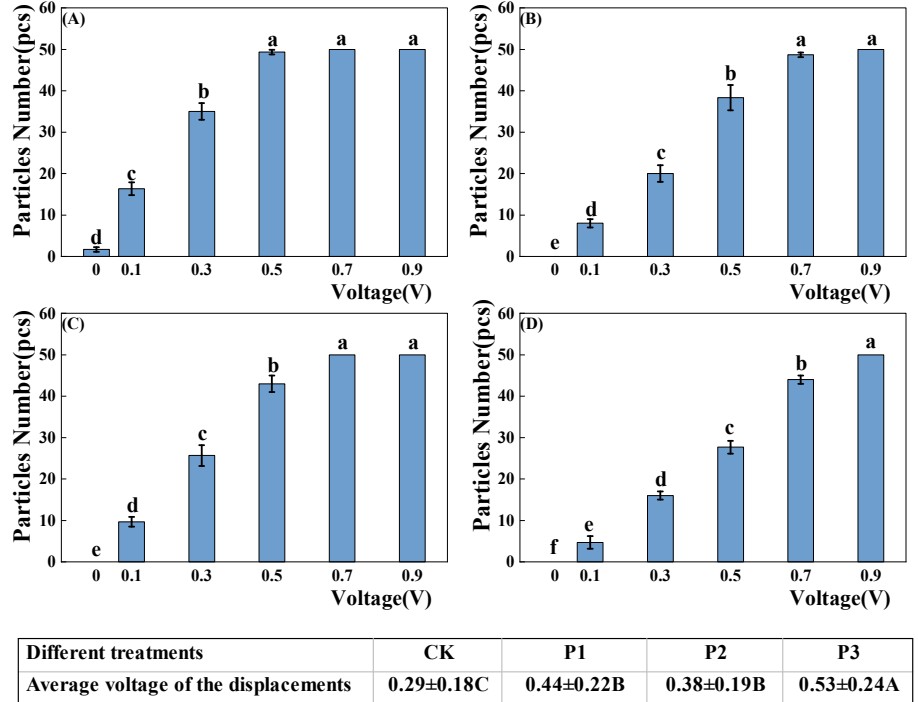

| Different treatments | CK | P1 | P2 | P3 |
|---|---|---|---|---|
| Average voltage of the displacements | 0.29±0.18C | 0.44±0.22B | 0.38±0.19B | 0.53±0.24A |

**Figure 3.** The number of soil particles displaced on mineral surfaces of indoor experiments. (**A**) The number of soil particles displaced at different voltages for the control group; (**B**) The number of soil particles displaced at different voltages for P1; (**C**) The number of soil particles displaced at different voltages for P2; and (**D**) The number of soil particles displaced at different voltages for P3. Different capital letters denote significant differences ($p < 0.05$) between various treatments based on a one-way analysis of variance and the Duncan test. Different lowercase letters denote significant differences ($p < 0.05$) between different voltages applied to the same bacteria.

Figure 4 displays the results of the pot experiment on the soil's adherence to minerals. The treatments that included bacteria significantly increased the soil's adherence to the minerals, with P3 having the most dramatic effect (Figure 4).

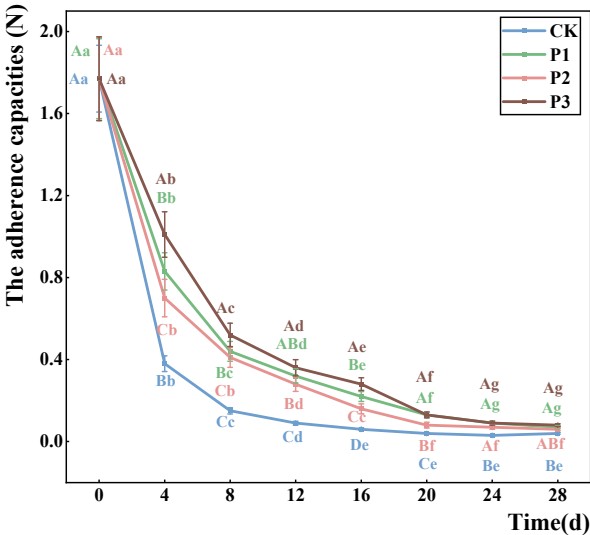

**Figure 4.** Soil's adherence to the minerals in the pot experiment. Different capital letters indicate a significant difference ($p < 0.05$) across treatments simultaneously based on a one-way analysis of variance and the Duncan test. Different lowercase letters denote significant differences ($p < 0.05$) between different timepoints applied to the same bacteria.

### 3.2. Soil Characteristics

Although the soil's moisture content gradually dropped, the treatments consistently added more bacteria than CK (Figure 5). The CK would reach a severe drought on day 4 (8% soil volumetric water content), whereas P1, P2, and P3 would reach a severe drought on days 8, 8, and 9, respectively.

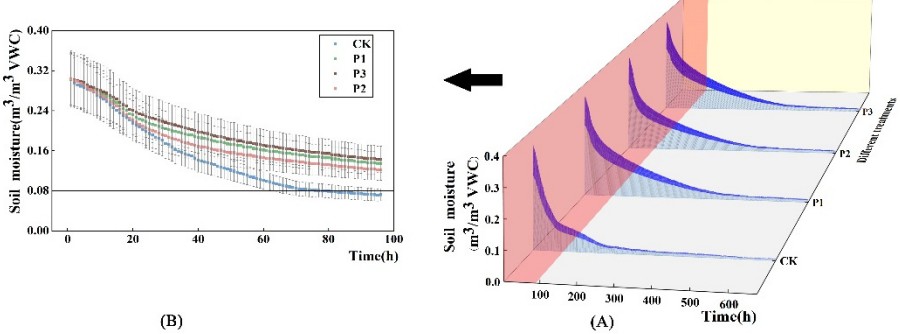

**Figure 5.** Soil moisture content with various soil bacteria. (**A**) Soil moisture content from 0 to 28 days; (**B**) Soil moisture content from 0 to 4 days.

The bacteria treatments caused the soil pH to be significantly lower than the CK (Figure 6). In contrast to CK, where there was no significant change in soil pH, P3's soil pH initially decreased, then increased over time, and finally stabilized at 7.0.

The amount of exchangeable metal (calcium, magnesium, sodium, and potassium) ions in the soil of the treatments with added bacteria gradually decreased over time, mostly between 0 and 16 days, in contrast to the CK, where there was no significant change in exchangeable metal ion concentration over time (Figure 7). Moreover, the P3 had the greatest impact compared to the added bacteria treatments.

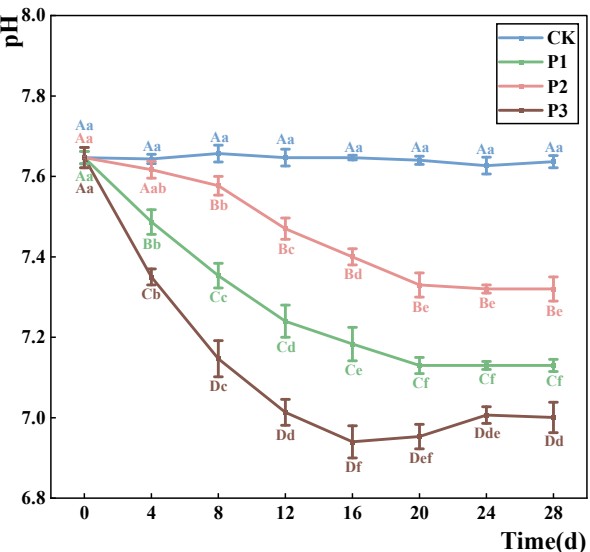

**Figure 6.** Soil pH with various soil bacteria. According to a one-way analysis of variance and the Duncan test, different capital letters indicate a significant difference (*p* < 0.05) across different treatments. Different lowercase letters denote significant differences (*p* < 0.05) between different timepoints applied to the same bacteria.

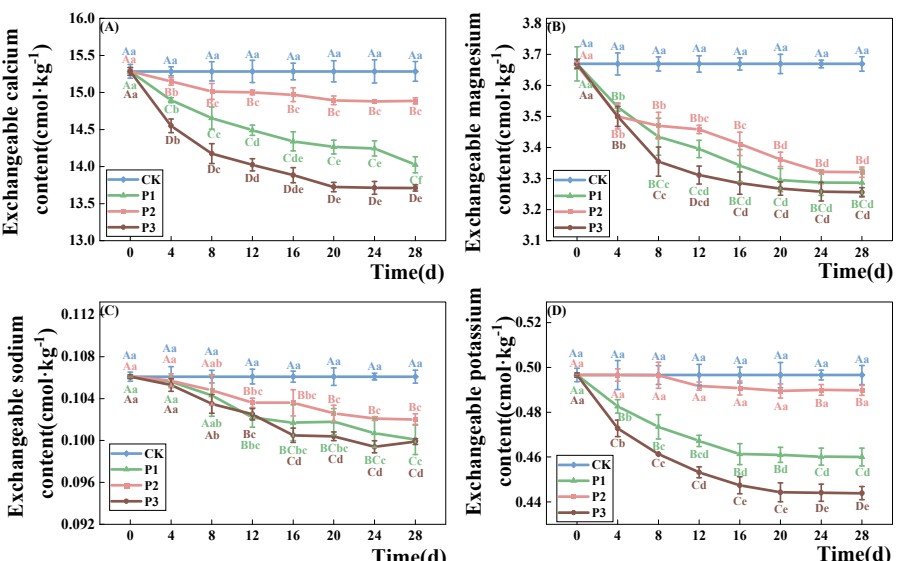

**Figure 7.** Soil exchangeable metal ion content with different soil bacteria. (**A**) Changes in exchangeable calcium ions in soil for various treatments; (**B**) Changes in exchangeable magnesium ions in soil for various treatments; (**C**) Changes in exchangeable sodium ions in soil for various treatments; (**D**) Changes in exchangeable potassium ions in soil for various treatments. Different capital letters denote significant differences (*p* < 0.05) between treatments simultaneously based on a one−way analysis of variance and the Duncan test. Different lowercase letters denote significant differences (*p* < 0.05) between different times applied to the same bacteria.

### 3.3. Effects of Soil Characteristics on the Adhesion of Soil to Minerals

There were highly significant positive correlations between soil adhesion to minerals, soil pH, soil moisture, and soil exchangeable magnesium and sodium ion concentrations. Significant positive correlations were found between the soil's adhesion to minerals and exchangeable calcium and potassium ion concentrations in the soil (Figure 8).

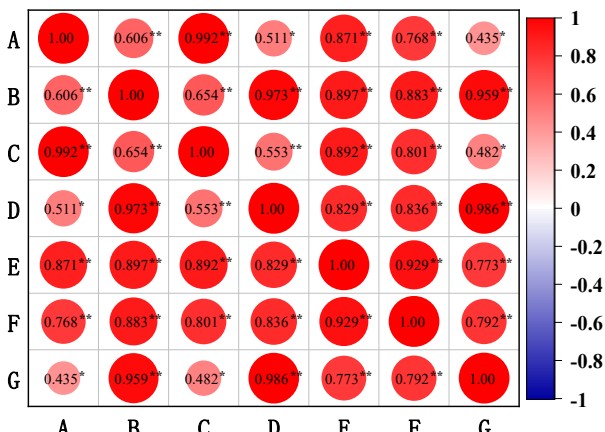

**Figure 8.** Correlation analysis of soil characteristics and soil adherence to minerals. The numbers in the circles represent the correlation coefficients between the parameters. ** and * indicate significant differences at $p < 0.01$ and $p < 0.05$, respectively. A, soil adhesion to minerals; B, soil pH; C, soil moisture content; D, soil exchangeable calcium ions; E, soil exchangeable magnesium ions; F, soil exchangeable sodium ions; G, soil exchangeable potassium ions.

The effect of soil characteristics on the adhesion of the soil to minerals was mainly achieved through soil moisture content (0.992), soil pH (−0.074), exchangeable magnesium ions (0.261) and sodium ions (0.034), of which soil moisture content had the greatest total effect (Figure 9; Table 1). Additionally, soil moisture content (0.901), soil pH (−0.271), and soil exchangeable magnesium ions (0.261) can all have a direct or indirect impact on soil adhesion to minerals (Figure 9). However, soil-exchangeable sodium ions (0.034) can only indirectly affect the soil's adhesion to minerals by affecting magnesium ions. The strongest indirect influence on soil adhesion is caused by soil pH (0.143), which primarily changes the soil's exchangeable metal ions, particularly sodium ions.

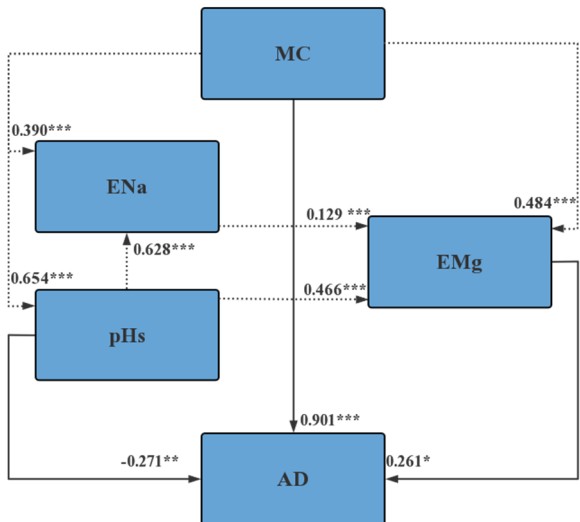

**Figure 9.** Structural equation model of soil characteristics affecting the soil adhesion to minerals. $p = 0.324 > 0.05$, GFI = 0.952 > 0.900, RMSEA < 0.08. Standardized path coefficients are shown as numbers on arrows. Solid lines indicate the direct influence of each parameter on the adhesion of soil to minerals, and dotted lines indicate the indirect influence of each parameter on the adhesion of soil to minerals. ***, **, and * indicate significant differences at $p < 0.001$, $p < 0.01$, and $p < 0.05$, respectively. AD: adhesion of soil to mineral; pHs: soil pH; MC: soil moisture content; EMg: soil exchangeable magnesium ions; and Ena: soil exchangeable sodium ions.

**Table 1.** The direct, indirect, and total effects of soil moisture content, pH, and exchangeable metal ions on soil adhesion to minerals based on structural equation models.

| Factors | Soil Moisture Content (MC) | Soil pH (pHs) | Exchangeable Sodium Ions (ENa) | Exchangeable Magnesium Ions (EMg) |
|---------|---------------------------|---------------|--------------------------------|-----------------------------------|
| Total | 0.992 | −0.074 | 0.034 | 0.261 |
| Direct | 0.901 | −0.271 | —— | 0.261 |
| Indirect | 0.091 | 0.143 | 0.034 | —— |

## 4. Discussion

China has recently intensified the construction of ecological civilization, and to implement the policy systems of carbon peaking, carbon neutrality, and "1 + N" [43–45], it is necessary to continually improve the scientific restoration of significant ecosystems and increase the capacity of ecosystems to store carbon [46,47]. Therefore, we extracted a range of efficient soil bacteria from the weathered surface of rocky slopes in Nanjing to treat rocky slopes scientifically and address the issue of difficulty in maintaining greening efficiency. However, most of the microorganisms used for ecological remediation in earlier investigations [48,49] were derived from soil or root systems. These isolated strains can be employed to enhance the traditional greening technology and to create a greening technique suitable for the fragile habitat of rocky slopes and, more importantly, possess Chinese characteristics, namely indigenous soil bacteria spraying greening.

In previous studies, we analyzed the tolerance of various strains using culture condition control experiments; we evaluated the impact of solubilization of various strains using fermentation culture tests, detecting the changes in the mixture's ion concentration and composition, etc. [33–35]. We used 16S rRNA to identify strains with improved behavior and filed patent applications for four strains [39,40,50,51]. Previous studies revealed that soil bacteria could enhance soil nutrients and promote plant growth [36,41]. It is important to note that during the test, we were surprised to discover that the treatment involving the addition of soil bacteria appeared to increase the soil's adhesion to minerals.

As a result, we selected bacteria (NL-7, NL-8, and NL-11) with high adaptability and improved applicability for our study into how soil bacteria affect soil adherence to minerals. The findings of this study adds to the knowledge in many fields, including the impact of mineral-solubilizing soil bacteria on soil adhesion capacity and the application of soil bacteria in slope ecological restoration. This bridged the knowledge gap on soil adhesion in ecological slope restoration and complemented the role of mineral-solubilizing soil bacteria in slope restoration. Permanent greening of rock slopes was made possible by a new greening method that combines native mineral-solubilizing soil bacteria with traditional spraying technology. This technique is safe, efficient, economical, and environmentally friendly.

Since soil bacteria were found to impact soil adhesion to minerals in our previous study, we hypothesized that NL-7, NL-8, and NL-11 helped improve soil adhesion to minerals. The findings of this study supported our hypothesis. When mineral-solubilizing soil bacteria were added, we discovered that the soil's adhesion to minerals was significantly improved (Figures 3 and 4). This could be attributed to the significant amounts of organic material that soil bacteria produce during their growth and metabolism [52–54], which alters the characteristics of the soil and minerals at the interface. It is possible that the P3 treatment (NL-11 supplementation) had a greater effect on promoting soil adherence to minerals than the P1 (NL-7 supplementation) and P2 (NL-8 supplementation) treatments because the addition of different bacteria produced distinct secreted chemicals and had varied effects on soil characteristics [52,55]. Moreover, we noted that the slope of the connection between adhesion to minerals and time varied across soil treatments with and without the adhesion of bacteria. This suggests that adding soil bacteria enhances soil's ability to adhere to minerals while delaying the decline of adhesion. This finding could help slopes become

more stable over time and better able to withstand extreme weather conditions by using innovative greening methods [56].

In this study, the treatments that included mineral-solubilizing soil bacteria, compared to the control, improved water retention in the soil (Figure 5). It was exciting to observe that, compared to the control, the P3 treatment with the best water retention may delay the onset of a severe drought in the soil by nearly six days. This would significantly increase the ability of slope ecosystems to withstand severe drought.

This can be attributed to several reasons. First, mineral-solubilizing soil bacteria improve soil quality by increasing the organic matter in the soil and improving the pore conditions and aggregate structure of the soil. Several studies have reported that bacteria positively impact soil organic matter and soil structure [57–59]. When there is a water shortage, the soil's surface particles shrink and the capillaries that connect them to the underlying aggregates break, which helps the soil retain water [60,61].

Second, the results of several studies have shown that a variety of *Bacillus* spp. can release extracellular polymeric substances (EPS) and form biofilms [62–66]. According to several studies, EPS has a high water absorption capacity and can even absorb up to 15–20 times its mass in water. This is mostly because soil absorbs moisture and has electrostatic and hydrogen bonds that act as a bonding mechanism. EPS will adhere to the dry surface without moisture and release water. Additionally, some studies have suggested that EPS can form fine filaments and two-dimensional structures in the soil to keep liquids connected; in the event of a water shortage, the network of two-dimensional structures will slow the diffusion of water vapor and hold the water in the soil [67,68].

In our study, we also discovered that adding mineral-solubilizing soil bacteria caused a significant decrease in soil pH compared to the control. This is consistent with findings of some previous studies [69–71]. The bacteria's ongoing carbon dioxide emission during daily activity, which combines with water to produce carbonic acid and dissociate hydrogen ions, may contribute to soil acidification. Additionally, the decrease in pH can be as a result of the numerous acidic chemicals produced, bacterial growth and metabolism, as well as the accelerated decomposition of organic materials by the bacteria [72,73].

Additionally, we found that mineral-solubilizing soil bacteria significantly reduce the number of exchangeable metal ions in the soil significantly more than the control (Figure 7). The structural equation modeling analysis results verified our hypothesized connection between this and the alteration in soil pH (Figure 9, Table 1). Large amounts of free hydrogen ions, which have a stronger adsorption capacity than calcium, magnesium, sodium, and potassium ions, are produced when soil bacteria are added. As a result, these soil nutrient ions are pushed out of the soil solution. The results of several earlier investigations also corroborate our hypothesis [74]. Hydrogen ions transport nutrient ions from the soil colloid into the soil solution and the plant through the root system. This may also explain the plant-promoting impact of mineral-solubilizing soil bacteria described in earlier investigations [75–77].

The results of structural equation modeling analysis demonstrated that soil moisture content, soil pH, and exchangeable magnesium and sodium ions could all affect how well soil adheres to minerals (Figure 9, Table 1). Notably, the results of the water tension theory, capillary theory, and ensemble theory proposed by Fountaine, Toyo Akiyama et al. are consistent with the observation that soil moisture content has the greatest direct and total positive effect on the adherence of soil to minerals [42,78–80]. Additionally, soil pH can have a direct negative effect on the adherence of soil to minerals, meaning that the stronger the adhesion of soil to minerals, the lower the soil pH, which is inconsistent with previous findings [12,14]. The significant amount of acid produced by adding soil bacteria and the water management conditions used in the experiment may have caused this disparity. In addition, this study discovered a significant indirect influence of soil pH on the adherence of soil to minerals, which was mostly achieved by influencing the exchangeable sodium and magnesium ions. Soil pH, in particular, has a significant effect on soil exchangeable

sodium ions. This is in compliance with Coulomb's law that higher valence ions have greater adsorption capacity than lower valence ions.

## 5. Conclusions

This study revealed that the three soil bacteria selected for this study with mineral-solubilization significantly increased soil adherence to minerals and reduced the rate at which soil adhesion decreased over time. Additionally, adding soil bacteria can change the soil's pH, improving soil fertility. It can also dissolve and release exchangeable metal ions (calcium, magnesium, sodium, and potassium) into the soil as nutrients, improving its capacity to hold water and delaying the occurrence of a soil water deficit. It is important to note that the P3 treatment (NL-11 added) significantly impacted the soil's adherence to minerals and soil characteristics.

(1) Soil moisture content has the greatest total and direct positive influence on the adherence of soil to minerals. This is because the soil moisture content is directly related to mineral-solubilizing soil bacteria. (2) The soil pH primarily affects the soil's exchangeable sodium ions, which in turn affects the adherence of soil to minerals. (3) Soil-exchangeable sodium ions can only somewhat influence soil adhesion to minerals by altering soil-exchangeable magnesium ions.

In conclusion, increasing the soil's ability to retain water can improve soil adherence to rock walls. This is done by introducing native, mineral-solubilizing soil bacteria into the substrate. With the help of a new modified soil greening method, slopes can be permanently greened while fostering a stable habitat for plant development and ecological restoration.

## 6. Patents

Jinchi Zhang, Guanglin Wang, Jiayao Zhuang, Qun Wang: An efficient limestone erosion bacterium Bacillus megaterium NL-7 and its application. CN103087953B; Jinchi Zhang, Guanglin Wang, Bo Zhang, Yanwen Wu: An efficient limestone erosion bacterium Bacillus thuringiensis NL-11 and its application. CN103087954B; Jinchi Zhang, Guanglin Wang, Li Wang, Bo Zhang: An efficient limestone erosion actinomycetes Streptomyces thermocarboxydus NL-1 and its application. CN103103151B; Guanglin Wang, Jinchi Zhang, Jie Lin, Rong Cao: An efficient limestone erosion fungus Gongronella butleri NL-15 and its application. CN103087926B.

**Author Contributions:** Conceptualization, J.Z. and L.W.; Methodology, L.W., X.T. and X.L.; validation, L.W. and X.T.; formal analysis, L.W. and X.T.; investigation, L.W.; resources, J.Z. and L.W.; data curation, L.W., X.T. and X.L.; writing—original draft preparation, L.W.; writing—review and editing, L.W., X.T., X.L. and J.Z.; funding acquisition, J.Z.; project administration, L.W. and X.L. All authors have read and agreed to the published version of the manuscript.

**Funding:** This research was funded by the Jiangsu Science and Technology Plan Project [BE2022420]; Postgraduate Research & Practice Innovation Program of Jiangsu Province [KYLX16_0864]; Innovation and Promotion of Forestry Science and Technology Program of Jiangsu Province [LYKJ (2021) 30]; Scientific Research Project of Baishanzu National Park [2021ZDLY01]; Priority Academic Program Development of Jiangsu Higher Education Institutions [PAPD].

**Data Availability Statement:** The data presented in this study are available upon request from the corresponding author.

**Conflicts of Interest:** The authors declare no conflict of interest.

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
