# Peer review of "Mineral-Solubilizing Soil Bacteria Permanently Green Rocky Slopes by Enhancing Soil Adhesion to the Surface of Rocky Slopes"

_forests, doi:10.3390/f13111820_

Round 1

Reviewer 1 Report

Review of the manuscript: Mineral-solubilizing soil bacteria achieve permanent greening of rocky slopes by enhancing soil adherence to the rocky slope surface

General comments: The manuscript is well written in English and reports the effect of mineral-solubilizing soil bacteria on soil adhesion to minerals; also investigate the effect of mineral-solubilizing soil bacteria on soil characteristics; as well the effect of soil characteristics on adhesion of soil to minerals under the action of mineral solubilizing soil bacteria. The introduction, methods, results are well documented. However, there are some specific comments for the methods and results section that should be resolved before publication.

Comments to authors:

METHODS:

It would be nice to include an image (photo) to see the experiment devices working in indoors and pot.

RESULTS:

Regarding this section:

The X-axis of graphics of figure 3 say VALTAGE should said “VOLTAGE”, likewise in Y-axis say NUMBE, should said “NUMBER”

The graphics of Figure 6 is not included in the manuscript

Lines 239 to 245 should move below title figure 9

Line 315 please define first what is EPS

Author Response

Dear Professor:

Thank you very much for giving us the positive and constructive comments and suggestions on our manuscript.

We have carefully studied your comments and have made every effort to revise our manuscript according to the comments and have marked the revisions in the paper.

We really appreciate your comments on our paper. A revised version has now been uploaded to the submission system for your consideration.

Thank you and best regards.

Sincerely,

Lingjian Wang.

Reviewer 2 Report

Review Report on the Manuscript Number: forests-1932677 Title: Mineral-solubilizing soil bacteria achieve permanent greening of rocky slopes by enhancing soil adherence to the rocky slope surface       

The manuscript investigates the effect of mineral-solubilizing soil bacteria on soil adhesion and soil characteristics and also studies the effect of soil characteristics on adhesion of soil to minerals under the action of mineral solubilizing soil bacteria. Generally, I think that the manuscript is well written and the topic seems to be appropriate for the journal of Forests. In the following, I suggest some possible improvements.

1.      I recommend summarize more accurate and representative keywords. Like the soil bacteria and soil moisture content, just general keywords which are common, are they appropriate to be key words for this paper?

2.      The Abstract is a very confusing for readers. The methods used and also objectives in this study should be presented together.

3.      Line 111, It would be good to add figure showing the geographical location and aerial map for your soil samples collected from Mufu mountain.

4.      Why did you choose these soil samples? According to what? Investigation or other researches?

5.      Line 140, I am not really convinced of drawing of Figure 1 shown in paper. It would be good to add a picture with high quality for readers.

6.      I suggest to add some new references to introduction section such as:

Effects of Mineral-Solubilizing Microorganisms on Root Growth, Soil Nutrient Content, and Enzyme Activities in the Rhizosphere Soil of Robinia pseudoacacia. Forests. Chong Li , Zhaohui Jia, Lu Zhai, Bo Zhang, Xiaonan Peng, Xin Liu and Jinchi Zhang…

7.      Line 299-300, It would be good to study and add these references for readers.

I suggest to study works:

Abdi, E., 2014. Effect of oriental beech root reinforcement on slope stability (Hyrcanian Forest, Iran). J. For. Sci. 60 (4), 166–173.

Parhizkar, M., Shabanpour, M., Lucas-Borja, M.E., Zema, 2021. Hydromulch roots reduce rill detachment capacity by overland flow in deforested hillslopes Journal of Hydrology 598: 126272.

8.      Line 325 to 329, here you present interesting results, you could explore and discuss these results further.

9.      Line 331-337, it would be better if you could report the actual and comparable results, so that your readers can see similar findings, if the experimental conditions were comparable to your study.

10.  Please explain relation and extension of results obtained from the study to natural conditions in larger scales (scaling).

11.  Please check the format of the references carefully.

Author Response

(The authors gave the same response as above.)
